# Robustness of Sparsely Distributed Representations to Adversarial Attacks in Deep Neural Networks

**DOI:** 10.3390/e25060933

**Published:** 2023-06-13

**Authors:** Nida Sardar, Sundas Khan, Arend Hintze, Priyanka Mehra

**Affiliations:** 1Department for MicroData Analytics, Dalarna University, 791 88 Falun, Sweden; v22nisar@du.se (N.S.); v22sukha@du.se (S.K.); ahz@du.se (A.H.); 2BEACON Center for the Study of Evolution in Action, Michigan State University, East Lansing, MI 48824, USA

**Keywords:** adversarial attacks, information smearedness, artificial neural networks, information relay, dropout, fast gradient sign method

## Abstract

Deep learning models have achieved an impressive performance in a variety of tasks, but they often suffer from overfitting and are vulnerable to adversarial attacks. Previous research has shown that dropout regularization is an effective technique that can improve model generalization and robustness. In this study, we investigate the impact of dropout regularization on the ability of neural networks to withstand adversarial attacks, as well as the degree of “functional smearing” between individual neurons in the network. Functional smearing in this context describes the phenomenon that a neuron or hidden state is involved in multiple functions at the same time. Our findings confirm that dropout regularization can enhance a network’s resistance to adversarial attacks, and this effect is only observable within a specific range of dropout probabilities. Furthermore, our study reveals that dropout regularization significantly increases the distribution of functional smearing across a wide range of dropout rates. However, it is the fraction of networks with lower levels of functional smearing that exhibit greater resilience against adversarial attacks. This suggests that, even though dropout improves robustness to fooling, one should instead try to decrease functional smearing.

## 1. Introduction

Neural networks are a highly effective tool in the field of artificial intelligence (AI) as they enable machines to learn and make predictions by analyzing extensive datasets. These networks are modeled on the complex structure and functionality of the human brain [1], featuring interconnected layers and nodes that can process intricate information. Consequently, neural networks can accomplish sophisticated tasks such as recognizing images [2], making predictions [3], enabling reinforcement learning [4,5], and providing the foundation for various generative technologies [6].

Minor alterations to input data can result in incorrect outputs from fully trained neural networks, a phenomenon known as fooling [7,8]. This can occur not only when these networks serve as classifiers but also to recurrent neural networks [9]. Several strategies, such as FGSM fooling [9], defensive distillation [10,11,12], and genetic algorithm-based image optimization [13], have effectively induced this fooling. FGSM fooling uses the neural network’s gradient information to produce adversarial perturbations, leading the network to make incorrect predictions. Defensive distillation trains a secondary network using pseudo-labels derived from a base network, and genetic algorithm-based image optimization generates adversarial examples that mislead the network’s predictions. Such techniques present substantial challenges to the broader implementation of neural networks, given their potential to manipulate systems such as self-driving cars using simple images or to deceive security cameras via clothing print [14].

Numerous countermeasures against fooling have been suggested, each demonstrating varying degrees of efficacy. Preprocessing or augmenting data are potential strategies [15,16,17,18], as is the detection of manipulated images [19]. Model compression [20] or diversification of the dataset [21] can also be employed. However, a comprehensive solution remains elusive.

The issue of fooling in neural networks underscores the necessity to enhance their robustness and generalization capabilities, thereby closing the gap between their performance and human abilities. Addressing these issues is of paramount importance for ensuring the dependable and secure deployment of neural networks in a variety of applications.

Interestingly, while artificial neural networks are inspired by natural brains, humans do not display the same vulnerability to fooling. Even in the face of severely degraded images, humans can still recognize objects, demonstrating a level of robustness that current neural networks do not possess. Furthermore, humans often demonstrate overgeneralization instead of overfitting, exhibiting an ability to generalize that extends beyond the training data [22].

The vulnerability of neural networks to fooling may be attributed to the differences in information processing compared with natural brains. In this study, we focus on the distribution of information within the neural network. Previous research has indicated that representations, or the information a neural network possesses about its environment, are dispersed throughout deep-learned networks [23,24,25,26]. In contrast, neural networks optimized using genetic algorithms do not exhibit this tendency and demonstrate greater robustness to noise [24,27,28]. Likewise, human brains, which have been “optimized” by evolution, employ distinct brain regions for computation [29]. We hypothesize that distributed representations are more susceptible to fooling than localized ones.

Dropout is a popular regularization technique that is commonly used to prevent overfitting and promote generalization in neural networks [30]. It has been suggested that dropout may also be effective in preventing fooling attacks, as it has been shown to mitigate FGSM fooling [31]. However, the impact of dropout on the neural network’s representation (that is, information about the environment, not features, see below) is not well understood. Dropout works by randomly setting a fraction of the weights in the neural network to zero, which forces each hidden unit to work with a randomly chosen subset of other units during training. This encourages each unit to develop useful features independently and become more robust to changes in the input. However, the hidden units within a layer still learn to perform different functions from one another, and they may specialize in differentiating between specific features [30]. As a result, the activations of the hidden units become sparse, even in the absence of sparsity-inducing regularizers. Thus, dropout may lead to the automatic development of sparse representations in the neural network.

In this context, the term “representations” refers to the number of features that a neural network extracts. It is not surprising that convolutional kernels can benefit from dropout when used correctly [32]. However, the type of representations that we are interested in are not just individual features, but rather the information that the network has about its environment and how that information flows through the network. By examining these representations, we can identify the functional modules that enable the network to perform its task by taking advantage of *relay information* [26] (See Section 2 for details). In other words, even if a network has a diverse set of features, it may not necessarily distribute them across different functions. For example, consider the MNIST task [33]. A network that has individual detectors for each numeral may have different functional modules for each numeral, each with several specific feature detectors. In contrast, a network with various sub-features that are necessary to differentiate between numerals would use all or most of its feature detectors for all numerals (see Figure 1 for an illustration).

The idea of having sparsely distributed functional modules is based on neuroscience research [29]. In natural brains, different functions are compartmentalized into modules, even though there are numerous feature detectors, such as in the retina [34]. In contrast, artificial neural networks trained using backpropagation recruit all weights into all functions simultaneously, preventing the emergence of functional modules, even though distinct feature detectors may be learned. While regularization and dropout techniques make networks more robust to overfitting and insensitive to fooling, they likely involve all weights and improve feature detection. However, the effect of these methods on creating functional modules and the degree of overlap between them is unknown.

The distinction between the enhanced feature detection facilitated by dropout and the functional modules identified by the information relay method becomes particularly salient in light of the concept of “Optimal Brain Damage” [35]. This idea entails calculating each weight’s contribution to the training error, which can be estimated by examining the diagonal values of the Hessian matrix. These diagonal values, associated with each weight (parameter) of the neural network, represent the second derivative of the loss function. If these values are low, it suggests that the weight only mildly contributes to classification and can be pruned. This implies that by making a network sparser—through having more weights become irrelevant (exhibit low curvature of the loss function)—the network’s robustness is increased. Interestingly, it has been demonstrated that dropout flattens the curvature of the loss function for each weight, thereby enabling more weights to be pruned [36,37,38,39]. Therefore, dropout effectively induces sparsity in the weight matrices, a phenomenon we will corroborate using our trained models.

However, it is crucial to note that the sparsity of the weight matrix does not necessarily equate to the functional modularity of the hidden layer states. Consider a neural network with a hidden and an output layer, defined by two weight matrices. The first matrix, *A*, might have the dimensions of i=784 input pixels by n=20 hidden nodes, while the second matrix, *B*, could have the dimensions of n=20 hidden nodes by o=10 output classes. Suppose we hypothesize two nodes, namely n=1 and n=2, forming a functional module necessary for controlling the output of the last class, o=9. In this case, all weights $A_{i,1,2}$ and $B_{1,2,9}$ contribute to that function. However, if we only consider the weights—as we would do when calculating the loss function curvatures using the Hessian matrix—we would be unable to identify which nodes, *n*, form functional modules, since the value of a single output node (e.g., o=9) depends on all weights in *A* and $B_{n,9}$. This comparison underscores the differences between the concepts of sparse weight matrices (and dropout-induced sparsity) and the functional modules we consider herein.

In this study, we explore the impact of various levels of dropout on networks trained to perform the MNIST task and investigate the correlation between dropout and the network’s ability to resist FGSM fooling attacks. We also assess the level of functional compartmentalization in the network by examining relay information. We compare those results with the effect that dropout has on making the curvature of the loss function for each weight flatter. Our goal is to investigate whether dropout can induce or prevent functional compartmentalization and to determine if networks with compartmentalized functional modules are more resilient to fooling attacks.

## 2. Materials and Methods

In this section, we detail the methodology used to create a neural network that can resist adversarial attacks. We start by describing the network’s architecture and the preparation of training and test data. We then explain how we incorporated dropout, a common regularization technique, into network training, and how we used the Fast Gradient Sign Method (FGSM) to generate fooled images for testing the network’s robustness. Additionally, we discuss the use of relay information to identify functional modules in the network.

### 2.1. Architecture of the Neural Network

To conduct this study, a neural network was designed with a specific architecture. The network consisted of three layers: an input layer with 784 neurons to handle the MNIST images, a hidden layer with 20 neurons, and an output layer with 10 neurons to represent the 10 different numeral classes. The hidden layer of the network also served as a dropout layer that incorporates dropout functionality when the probability is greater than zero [30]. The highest output value neuron was used to indicate the class of the input through the use of softmax. The threshold function used in the hidden layer was a hyperbolic tangent, while the output layer used a sigmoid.

In theory, provided that the size of the hidden layer is sufficiently large [40,41], the network should be capable of performing the MNIST task. However, factors such as smearing, fooling, the effect of dropout, and the extent of overfitting (stability) could be influenced by the size of the network. Therefore, a range of network sizes was examined (as detailed in the Section 4), and an optimal size of 20 was selected for this experiment.

### 2.2. MNIST Dataset

The MNIST dataset is widely recognized as the “Hello world!” example for deep learning and comprises 28×28 pixel images of handwritten digits ranging from 0 to 9. It contains 60,000 images for training and 10,000 images for testing the neural network. To prepare the dataset for the dense network, the 28×28 pixel images were converted into a vector of 784 entries by normalizing the pixel values, which lie within the range of [0.0, 1.0].

### 2.3. Training Neural Networks

To investigate the effect of dropout on the network’s robustness, 100 neural networks were trained for each dropout value using the default Kaiming method [42] from PyTorch to seed the networks with random weights. The Adam optimizer [43] was used with the mean squared error (MSE) loss function. The networks were trained until they reached a minimum accuracy of 96%.

### 2.4. Fooling Using Fast Gradient Signed Method

The Fast Gradient Sign Method (FGSM) is a widely used technique for generating adversarial images capable of tricking image recognition systems. The method involves introducing a small perturbation to the original image, causing it to be misclassified by the system. To accomplish this, a forward and backward pass is performed, similar to normal classification, but, instead of updating weights to improve classification accuracy, the perturbation required to misclassify the image is calculated. The magnitude of the perturbation is controlled by a parameter called ϵ, which can be set by the researcher. If ϵ is set to zero, no perturbation is applied, and, as ϵ increases, the image is increasingly distorted. An illustration of the FGSM process can be found in Figure 2, which shows the original image, the perturbation, and the resulting fooled image.

To evaluate a network’s resistance to fooling, we test its classification accuracy for varying levels of ϵ (ranging from 0.0 to 0.3) using the FGSM technique. Higher accuracy for larger ϵ values indicates higher robustness against fooling. The mean accuracy is calculated for each ϵ value by sampling over (0,0.05,0.1,0.15,0.2,0.25,0.3), and all 10,000 test images are used to generate fooled images at each ϵ level.

### 2.5. Relay Information, Greedy Algorithm, and Functional Modules

An artificial neural network can be viewed as an information-theoretic channel. When perfectly trained, the information entering the network, such as input images of numerals, correlates flawlessly with the output classes. When not trained perfectly, this correlation becomes weaker. Channel information quantifies this correlation using entropy measures, and can thus determine how much information “flows” through the network. Nevertheless, information is transmitted from one layer to another, with different hidden nodes potentially conveying distinct portions of the information. Relay information, $I_{R}$, ref. [26]] measures the amount of information passing through a subset of nodes in the hidden layer that is not transmitted through the remaining nodes within the same layer. The calculation of $I_{R}$ (as defined in Equation (Equation 1)) necessitates the measurement of four random variables (refer to Figure 3 panels A and B):$X_{in}$—the input classes;$X_{out}$—the classification results;$Y_{R}$—the hidden states of the subset deemed responsible for relaying the information;$Y_{0}$—the residual set of hidden states not incorporated in $Y_{R}$.
(1)$$I_{R}=H\left(X_{\mathrm{in}};X_{\mathrm{out}};Y_{R}|Y_{0}\right);,$$

It should be noted that $I_{R}$ is not only an indicator of how functionally relevant a set of nodes is, but also how much information passes through that set compared with the rest of the nodes in the same layer. Sets of nodes with high $I_{R}$ are therefore highly functionally relevant. However, given that there are $2^{N}$ possible bi-partitions for *N* nodes, it can be challenging to identify which set of nodes is essential. Fortunately, a greedy algorithm can be used to address this issue. The algorithm first computes $I_{R}$ for all hidden nodes, and then calculates $I_{R}$ for each set obtained by removing one node at a time. The node whose removal has the least impact on $I_{R}$ is considered the least important and is removed. This process is repeated until all nodes have been removed. The output is a sequence of nodes in ascending order of importance in relaying information from inputs to outputs. This calculation is repeated for every output class, resulting in aggregated information, $I_{A}$, for each node and each function (see Figure 3 panel C as an illustration). This information is displayed in a matrix, *M*, with dimensions C×N, where *C* is the number of classes the neural network classifies and *N* is the number of nodes. This matrix, called the functional association matrix, shows which nodes are associated with which function (see Figure 1 panel B for an illustration).

### 2.6. Smearedness

In this context, “smearedness” refers to how the functionality of a neural network is distributed across its nodes and the classes it recognizes. If sets of nodes are organized into independent modules that contribute to different functions, the distribution is considered sparse. Conversely, if nodes overlap and contribute to multiple functions, the distribution is considered “smeared”. To quantify the smearing of each node, we use the *M* matrix, which contains the aggregated information, $I_{A}$, for each node and each concept. The smearing of each node can be calculated based on this matrix, as described in Hintze et al., 2018 [23].

## 3. Hessian Matrix and Sparsity Induced by Dropout

The diagonal of the Hessian matrix reflects the curvature of the loss function. Its calculation, while computationally demanding, is well-defined [44,45]. To ascertain the Hessian matrix, one can either use random inputs and outputs, or samples from training or testing data. The function employed in this study (the hessian from PyTorch in conjunction with cross entropy as a loss function) computes the Jacobian of the Jacobian matrix for all parameters (weights) of the network. This yields the Hessian matrix, the diagonal of which delineates the curvature of the loss function for each parameter, given the input and output data.

The resulting curvatures provide an approximate measure of the influence of each weight change on the loss. When using random inputs, we obtain a broad understanding of how much each weight generally impacts outputs. However, when using training or test data, we can ascertain how much each weight is specifically implicated in the classification task.

In this context, we are interested in discerning the contribution of each weight to each possible classification function. Therefore, the Hessian matrix is calculated for 100 randomly selected images from each of the 10 numerals independently. This generates one vector extracted from the Hessian diagonal for each numeral class. Each element of such a vector illustrates the extent to which its associated weight contributes to the loss calculated for a single class. A low value suggests that the weight is not crucial for the classification of the specific class for which the Hessian matrix is calculated. A high value, on the other hand, implies strong involvement in the classification of that class.

For each trained neural network, we compute the correlation coefficient, $H_{c}$, between all those 10 vectors. A high correlation signifies that weights contribute in a similar way to all functions, whereas a low correlation suggests that weights contribute differently to all 10 classes. In other words, a high correlation indicates a more sparse association between weights and functions.

## 4. Results

For all experiments, the MNIST handwritten numeral dataset was utilized. This dataset contains 60,000 grayscale training images and 10,000 for testing. The same dataset has been used to verify the relay information method previously [26]; hence, it is a reliable basis for the assessment of representational smearing. Although other fooling methods are available, we concentrated our efforts on FGSM fooling [9] in this study.

Potential dependencies of dropout, smearing, and robustness to fooling on the size of the network, particularly the width of the hidden layer, were considered. Therefore, we trained neural networks with varying widths of hidden layers, ranging from 10 to 30, in increments of 2, for 20 epochs. The test and training accuracies were recorded to check for potential overfitting effects (refer to Figure 4A). Given that dropout is intended to prevent overfitting, it is crucial to ensure that the potential effects we observe are not simply due to this confounding factor. Otherwise, possible effects on fooling might arise from the prevention of overfitting and not making networks generalize more. Over the range of network sizes tested, we found that overfitting does not play a significant role. Moreover, networks with sizes below 20 demonstrate improved accuracy when nodes are added, but this effect becomes less pronounced after 20 hidden nodes (20 nodes is arguably at the inflection point), suggesting that a hidden layer of 20 nodes may indeed be a good choice.

While assessing the impact of network size on fooling robustness (see Figure 4B, black line), we found that the optimal robustness is achieved with networks of size 18, with an insignificant decrease observed for networks of size 20. There seems to be a trend in the reduction of robustness for larger networks. This further supports the choice of networks with a hidden layer comprising 20 nodes, where the highest fooling robustness is observed.

We also measured the smearedness for networks of varying hidden layer sizes, and it decreases as networks become larger (see Figure 4B, red line). It is important to note that the smearedness measure, which computes partial sums over a matrix, is confounded by the size of the matrix. Hence, measuring different values for different sizes is to be expected. Consequently, comparing the degree of smearedness between networks of different sizes may not be feasible in the first place. Nevertheless, we present these results here as a reference.

When we train the neural network on the MNIST dataset with varying dropout rates, we expect that it will affect the network’s robustness to FGSM fooling. As has been shown previously, dropout can have a positive effect on robustness, which we can confirm in our experiments with a dropout probability of $p_{dropout}=0.05$ (see Figure 5). However, this effect is small and limited to a particular range of dropout probabilities. This may be due to the specific dataset and hyperparameters used. Nonetheless, our results confirm previous findings [46]. On the other hand, as the dropout rate increases, the network’s robustness to fooling decreases.

While dropout is designed to mitigate overfitting, its effectiveness might vary depending on different dropout probabilities. Success or failure in preventing overfitting could confound robustness to fooling or smearing. Similarly, to rule out that dropout has different effects on the 10 numeral classes—for example, it might improve one class while being ineffective on another—we compared the impact of dropout on overfitting for each of the 10 numerals separately.

Although there were minor performance differences among the 10 numeral classes (see Figure 6), we found this effect to be minimal. We also observed that the difference between training and testing accuracy was consistently minimal (at most around 1%). Therefore, we anticipated that our other results, which are related to fooling or smearing and are caused by different degrees of dropout, would not depend on whether dropout has caused networks to behave differently with respect to different classes or overfitting/generalization.

Next, for all trained networks, the $I_{R}$ and the functional association matrices were computed and the degree of functional smearing was determined. Interestingly, we found dropout to increase significantly, smearing over a wide range of dropout rates (see Figure 7 panel A). This is a remarkable finding because it confirms our initial intuition, that dropout, while improving feature detection, still does not lead to a better function separation, but instead can drive functional smearing. We also found that, again as expected, robustness to fooling negatively correlates with functional smearing (see Figure 7 panel B), and that less smeared networks are indeed more robust to fooling. This is analogous to our initial argument that natural brains seem to not suffer from fooling as much as their artificial counterparts do, possibly due to them arranging functions into modules. Here we confirm that this principle is applied to artificial neural networks. Observing that the inverse conclusion might not apply, our results do not show that the robustness to fooling in natural neural networks comes from being modular. Our results do not contradict this but show the correlation for the networks trained here.

### The Effect of Dropout on the Slope of the Loss Function

We previously proposed that dropout, while successful at preventing overfitting, does not serve as a good predictor for fooling robustness and, in fact, encourages functional “smearing”. Unfortunately, functional “smearing” negatively correlates with fooling robustness. Therefore, the utility of dropout in preventing fooling is quite limited. At the same time, dropout improves feature detection and reduces the significance of individual weights, thereby allowing for network sparsity through pruning. These two ideas—of dropout not preventing fooling effectively while also driving functional sparsity—seem to be contradictory.

As explained in the Introduction, dropout has two roles: improving feature detectors and enhancing the sparsity of weight matrices. On the other hand, functional modules exist at the level of the hidden layer nodes, which are defined by the information relay method. We have demonstrated (see Figure 7) that less smeared (more modular) functional assignment correlates with more robust networks.

However, we can also assess how much each weight contributes to each function, and how much dropout influences this contribution through an alternate method. The diagonal of the Hessian matrix, given the set of inputs and outputs to the network, indicates the slope of the loss function for each weight (parameter) of a neural network. Particularly, low values on this diagonal signify weights that do not contribute significantly to the measured function. Thus, we computed the Hessian values for all networks trained with different dropout rates. However, the Hessian matrix for each model was computed independently for each of the 10 numeral classes, allowing us to evaluate the involvement of each weight in each of the 10 numeral classes independently (see Section 2).

By correlating these vectors with each other ($H_{c}$), we can measure the extent to which weights contribute to the same or different functions. High values of $H_{c}$ indicate that weights are involved in multiple functions simultaneously, while low values indicate a clear separation (sparsity) of functions for each weight.

Our results showed that, as previously described, dropout promotes the separability of features (see Figure 8A). Simultaneously, this separation (low $H_{c}$) makes networks more vulnerable to fooling (see Figure 8B). Finally, we found only a very weak correlation between $H_{c}$ and functional smearing. These results reinforce our earlier findings (Figure 5 and Figure 7) that dropout is a weak predictor or driver of fooling robustness, despite improving feature separation. These findings also support the notion that functional modularization, as defined by the information relay method, is a better predictor for fooling robustness.

## 5. Discussion

Dropout is a widely employed technique for mitigating overfitting in deep learning, and, as a result, it is also utilized to counteract fooling, such as the FGSM fooling examined in this study. Intriguingly, dropout has been found to enhance feature detection, which is thought to contribute to network robustness. In this research, we investigated the impact of dropout on both fooling and functional modularization. Functional modules, defined here as sets of nodes with high relay information, expand the concept of features by determining which network components contribute to the identification of each class. Our findings reveal that, while dropout improves fooling robustness as anticipated, it also influences functional modularization. However, rather than fostering more modular systems, it results in more functionally dispersed ones. Paradoxically, the systems exhibiting less dispersion are more resistant to fooling.

In our previous studies [24,25,47], we demonstrated the significant influence of genetic algorithms on the distribution of representations, elucidating that evolved neural networks exhibit less information smearing. Despite our advancements, we have not identified a backpropagation method that exhibits similar effects, possibly due to the novelty of the concept of representational smearing. Notably, even dropout, with its ability of modulating feature detection, appears to induce less functional modularity in network representations. This observation, therefore, presents an new avenue for future research: the exploration for a backpropagation method capable of condensing representations.

In this study, we exclusively used the MNIST dataset to demonstrate the effect, and employed a single network architecture with one hidden layer. Future research should explore more complex datasets, networks, and training regimes. Nevertheless, our results, based on these data and network structure, suggest that stricter functional modularization could potentially create networks with significantly enhanced resistance to fooling, although the variation induced by dropout does not offer this functionality.

Alternative approaches, such as elastic weight consolidation (EWC) [47], also aim to improve network modularization in the hopes of achieving better generalization, fooling robustness, or overcoming catastrophic forgetting. Examining the impact of these methods on the functional modules defined in this study may yield further insights for developing more robust functional modules in the future.

## Figures and Tables

**Figure 1 entropy-25-00933-f001:**
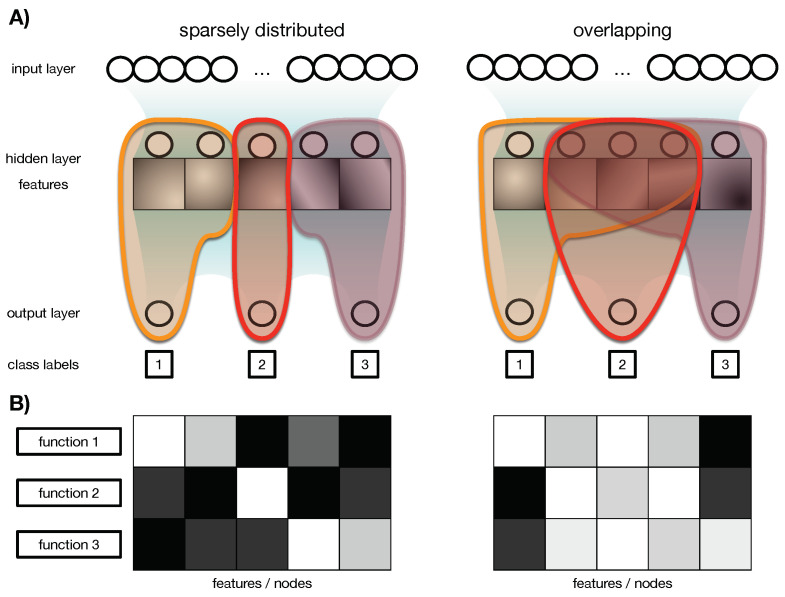
Illustration of the difference between separate feature detectors and sparsely distributed functions. Panel (**A**): On the top, the input layer of a neural network is connected to the hidden layer and the output layer. Here we assume a classification task to distinguish three numerals: 1, 2, and 3. On the left side, we have 5 feature detectors, which can be convolutional kernels or just nodes in the hidden layer detecting a feature. The orange, red, and purple sections depict how those detectors can contribute to the outputs. On the left, we find three distinct modules, one for each numeral, but they do not rely on the same features. On the right, we find overlapping modules, where each of the functional modules involves features other modules also rely on. Panel (**B**) shows an illustration of a functional association matrix. Squares in white show high functional involvements between the categorized class and the nodes of the hidden layer (feature detectors, respectively), and black squares indicate no functional involvement between a node and a function.

**Figure 2 entropy-25-00933-f002:**
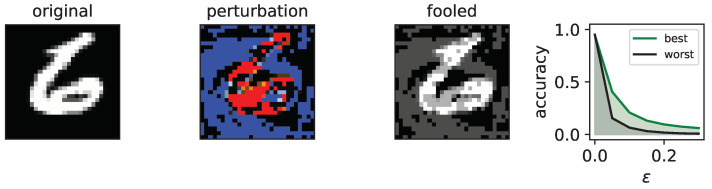
The figure illustrates FGSM fooling. The three images of the numeral 6 as *original* data, the *perturbation* computed by the FGSM fooling in red positive in blue negative values, and the resulting *fooled* image where the perturbation is applied to the original image. The plot on the right further showcases the classification accuracy across different values of ϵ, which represents the magnitude of the perturbation ranging from [0.0,0.3]. The data from the network that could be fooled the easiest are shown as *worst* in black, and the most robust network is shown in green as *best*.

**Figure 3 entropy-25-00933-f003:**
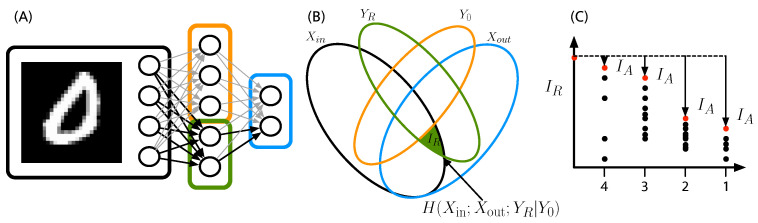
Illustration of relay information. Panel (**A**) depicts a neural network with an input image showing a 0 from the MNIST dataset. Information propagates through the network, represented by the gray or black arrows (where black arrows are assumed relevant), leading to the green circled hidden nodes being the relevant relays ($Y_{R}$) and the irrelevant nodes being circled orange ($Y_{0}$). Panel (**B**) displays $I_{R}=H\left(X_{\mathrm{in}};X_{\mathrm{out}};Y_{R}|Y_{0}\right)$ as the green surface in the information-theoretic Venn diagram, which also encompasses the input and output variables $X_{in}$ and $X_{out}$. Panel (**C**) details how the $I_{R}$ for each set of nodes of different sizes is calculated (points). The greedy algorithm commences at the largest set size and continues to successively remove the least significant node, yielding a sequence of nodes of increasing importance (red dots). The $I_{A}$ for each node is represented by the information loss upon its removal.

**Figure 4 entropy-25-00933-f004:**
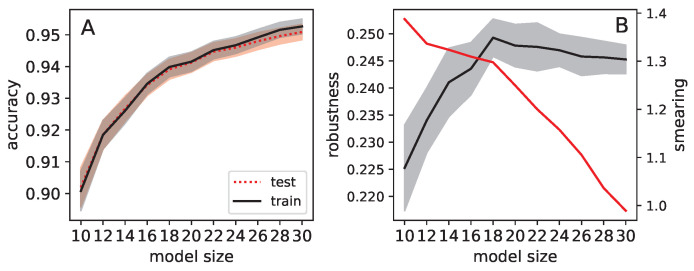
Properties of networks with different hidden layer sizes (10 to 30 in increments of 2) trained on the MNIST task. Results from 30 replicates, each trained for 20 epochs, are shown. In panel (**A**), training accuracy is in black, and test accuracy is in red. The shadows indicate the 95% confidence intervals. In panel (**B**), fooling robustness is in black (left *Y* axis), and smearing is in red (right *Y* axis). Again, shadows indicate 95% confidence intervals.

**Figure 5 entropy-25-00933-f005:**
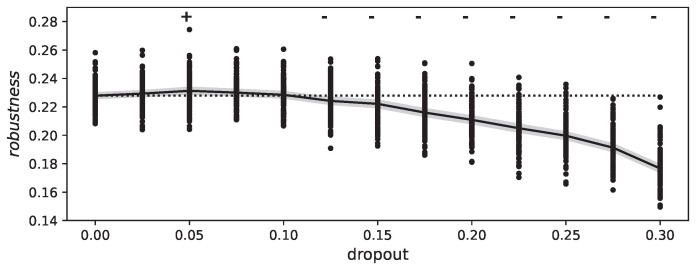
This figure shows the effect of varying dropout values on the robustness of a deep neural network to adversarial attacks. The *y*-axis represents the mean accuracy of the model after an attack, while the *x*-axis represents the dropout value used during training. The figure suggests that the model is more robust to attacks at dropout values between 0.035 and 0.05. The Mann–Whitney U test was used to analyze the significance of the results: + indicate significantly higher robustness, - for significantly lower (p<0.05).

**Figure 6 entropy-25-00933-f006:**
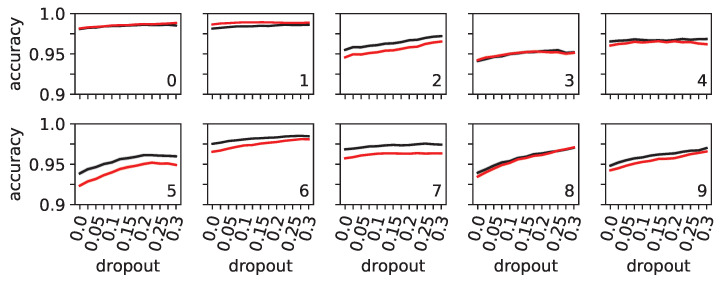
Mean training (black) and testing (red) accuracy of neural networks trained with different levels of dropout (*x* axis). The same networks used in Figure 5 were used here. Shadows in black and red behind the solid lines (almost negligible) indicate 95% confidence intervals. Each panel shows the accuracy for each of the ten numerals tested. When measuring the accuracy for a numeral, the dataset contained 500 images of the indicated numeral, and 50 images of other numerals.

**Figure 7 entropy-25-00933-f007:**
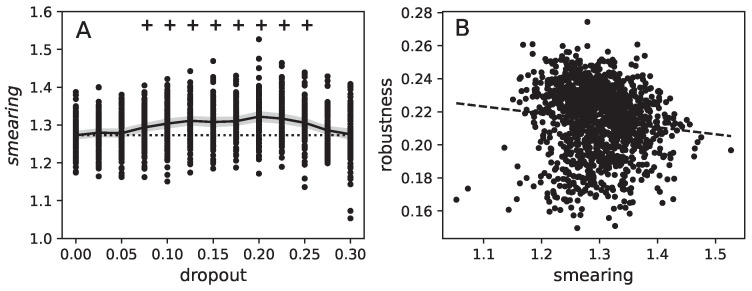
Correlation between functional smearing, dropout, and robustness to fooling. Panel (**A**) shows the average functional smearing over the dropout rate (black line) measured for all trained networks (individual results for each network are shown as black dots). The gray shadow shows the 95% confidence intervals. The + indicates where the distribution of functional smeardness at different dropout levels is significantly larger than the distribution when no dropout is applied, and the dashed line shows the average (*p*-value of a Mann–Whitney U test 0.001 for dropout 0.075, in the other cases $p<0.1\times 10^{-5}$). Panel (**B**) shows the correlation between the robustness to fooling (*y*-axis) over the degree of smearing (*x*-axis), again for all trained neural networks. The dashed line shows a linear fit to the data, and the significance of the correlation is $p<0.1\times 10^{-9}$.

**Figure 8 entropy-25-00933-f008:**
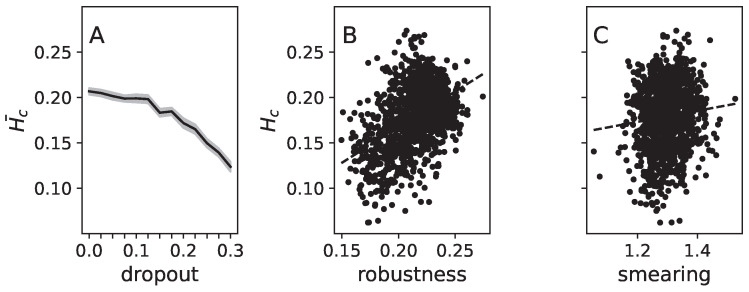
Correlation coefficient between Hessian matrix diagonals ($H_{c}$) obtained from models tested on the 10 different MNIST numeral categories. Panel (**A**) presents the mean $\bar{H_{c}}$ averaged over all 100 trained models (black line) for each degree of dropout (*x*-axis). The gray shadow indicates the 95% confidence intervals. Panel (**B**) displays the correlation between $H_{c}$ and robustness for all trained networks as a scatter plot. The dashed line shows a linear fit to the data, and the correlation coefficient is 0.42 with a *p*-value virtually zero (<$0.9\times 10^{-50}$). Panel (**C**) illustrates the correlation between $H_{c}$ (again for all tested models) and their functional smearing (*x*-axis). The dashed line shows a linear fit to the data, and the correlation coefficient is 0.08 with a *p*-value of 0.003, indicating a very weak correlation.

## Data Availability

The code to generate and analyze the data can be found here: https://github.com/Hintzelab/Robustness-of-Sparsely-Distributed-Representations-to-Adversarial-Attacks-in-Deep-Neural-Networks.

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
