# Peer review of "Robustness of Sparsely Distributed Representations to Adversarial Attacks in Deep Neural Networks"

_entropy, 2023, doi:10.3390/e25060933_

Round 1

Reviewer 1 Report

Suggestions for authors regarding the reviewed manuscript are included in the appendix below.

Author Response

Please see the attached file, Thanks. 

Reviewer 2 Report

The article is devoted to the actual problem of protecting neural networks from attacks and the study of the dropout rate on network security. However, it seemed to me that some points are not sufficiently considered in detail in this study, I would like to draw the attention of the authors to them:

1. The annotation should be expanded with a description of quantitative results, to what extent the stability of the network can be improved after the use of dropout.

2. The introduction section needs to be improved, since alternative, existing approaches for protecting a neural network from attacks (for example, distillation of neural networks and data) are not considered in sufficient detail.

3. Figure 2 does not contain a signature.

4. The Materials and Methods section needs to be improved. Firstly, it contains a number of information, obviously taken from other studies (description of the MNIST dataset, FGSM method), which should be included in the introduction or in the description of the experiment in the Results sections. Next, I would like to see a more detailed description of the IR and IA variables.

5. The results section should also be expanded. In the paper, the authors investigate various dropout options, the effect of this parameter on the strength of the network and smearing. I would like to see the results of practical research, conditionally before and after, when improving the dropout parameter would lead to an increase in the resistance of the neural network to attacks on specific examples.

Thus, my conclusion on the work is that it is certainly interesting, but it leaves a feeling unfinished, since many interesting ideas and research are planned by the authors to be carried out in future works.

Round 2

Reviewer 1 Report

I would like to thank the authors for their constructive response to the suggestions formulated in the review. In my opinion, the currently introduced additions to the text of the work significantly increase its substantive value. In my opinion, the work in its current form can be published.

Reviewer 2 Report

I have studied the answers to my questions, as well as the changes made. The authors have done a lot of work to improve the article and answered all the questions in sufficient detail.   The current version can be recommended for publication.